



# The value of ultra-detailed survey data for an improved flood damage modelling with explicit input data uncertainty treatment: INSYDE 2.0

Mario Di Bacco[1], Daniela Molinari[2], Anna Rita Scorzini[3]

[1] Department of Civil and Environmental Engineering, University of Florence, 50139 Firenze, Italy
[2] Department of Civil and Environmental Engineering, Politecnico di Milano, 20133 Milano, Italy
[3] Department of Civil, Environmental and Architectural Engineering, , University of L'Aquila, 67100 L'Aquila, Italy

*Correspondence to*: Anna Rita Scorzini (annarita.scorzini@univaq.it)

**Abstract.** Accurate flood damage modelling is essential to estimate the potential impact of floods and to develop effective mitigation strategies. However, flood damage models rely on diverse sources of hazard, exposure and vulnerability data, which are often incomplete, inconsistent, or totally missing. These issues with data quality or availability introduce uncertainties in the modelling process and affect the final risk estimations. In this study, we present INSYDE 2.0, a flood damage modelling tool that integrates ultra-detailed survey and desk-based data for an enhanced reliability and informativeness of flood damage predictions, including an explicit representation of the effect of uncertainties arising from an incomplete knowledge on the variables characterizing the system under investigation.

## 1 Introduction

In recent years, a policy shift from a mere hazard control to a more holistic flood risk management has steadily increased the demand for reliable quantitative flood risk assessment methodologies (Sayers et al. 2002; Merz et al., 2010). In this scenario, despite the significant advancements achieved in flood damage modelling over the past decade, the application of developed tools in practical decision-making for flood risk management has been limited, mainly because of concerns on modelling uncertainties affecting the results of loss estimations (Morgan et al., 1990; Apel et al., 2008).

Uncertainty, arising from an incomplete knowledge of the system under investigation, in terms of input data and/or model assumptions, could be reduced by enhancing model complexity (i.e., better representation of modelled mechanisms) and/or by using high quality input data (Wagenaar et al., 2016). In this regard, recent literature demonstrated that multi-variable flood damage models not only outperform simpler (stage-damage) functions (Schröter et al., 2014; Wagenaar et al., 2017; Amadio et al., 2019), but they also provide ancillary advantages, such as the identification of the most important damage explanatory variables (useful, for instance, to guide interventions for improving building resilience), and the possibility, for probabilistic models, of explicitly handling uncertainty into the modelling framework, thus supporting comprehensive and informative damage assessments (Morgan et al., 1990; Rözer et al., 2019; Zarekarizi et al., 2020). However, practical



constraints, such as budget, operational timelines, computational efforts, as well as issues on data quality and availability, often hinder the actual implementation of such models at large (e.g., river basin) scale, with the consequent risk of providing decision-makers with a limited perspective on potential damage scenarios (Pappenberger and Beven, 2006; Merz et al.,
2008; Wagenaar et al., 2016; Albano et al., 2018; Zarekarizi et al., 2020; Razavi et al., 2021).

With specific reference to data, for the case of residential buildings, literature has pointed out that several features characterizing both the event (e.g., water depth, flow velocity, inundation duration, debris and contamination loads) and the exposed object (e.g., material and construction type, age and finishing quality of the building, in addition to its geometrical parameters and more micro-scale characteristics) affect the flood damaging process (Penning-Rowsell et al., 2005; Dottori et
al., 2016; Wagenaar et al., 2016; Mohor et al., 2020; Nofal et al., 2020; Malgwi et al., 2021; Paulik et al., 2022). Hence, to ensure reliable flood damage assessments, there is a critical need to gather detailed data on vulnerability and exposure features, including relevant ancillary information, in order to achieve a thorough characterization of the buildings. Unfortunately, the availability and reliability of required data is generally low, especially for large-scale applications (Papathoma-Köhle et al., 2007; Schröter et al., 2018; Bhuyan et al., 2022; Velez et al., 2022).

To tackle this issue, a few existing tools have been designed to adapt to actual available knowledge on hazard and building features: an example is represented by INSYDE (Dottori et al., 2016), which is a synthetic (i.e., based on "what-if" analysis) multi-variable flood damage model for residential buildings, capable of handling missing input data by assigning them specific default values typical for the country/region of implementation (Dottori et al., 2016; Molinari et al., 2017; Scorzini et al., 2022). Still, the use of this approach could lead to biased results since missing and known inputs are treated as
equivalent when the former are set to their corresponding built-in defaults. Such a problem could be overcome by considering probabilistic distributions of unknown input data, within a Monte-Carlo approach. In this case, representative empirical distributions for the variables at stake are required to both consider the local nature of flood damage mechanisms and to obtain indicative and reliable uncertainty bounds (Cammerer et al., 2013; Wagenaar et al., 2018; Sairam et al., 2019; Scorzini et al., 2021, 2022). These distributions should be derived based on the actual characteristics of the analyzed area,
which implies the availability of building inventories and/or the possibility of conducting ad hoc surveys. However, the commonly poor availability of specific databases (especially regarding very micro-scale building attributes, such as the elevation of the first floor from ground level or the perimeter of internal walls), on the one hand, and the time-consuming operation of carrying out surveys, on the other hand, are currently the main obstacles to thorough analyses of model's sensitivities to uncertainties stemming from input data.

The divergent needs of balancing modelling costs and informative results (Di Bacco et al., 2023; Sieg et al., 2023) then pose two questions concerning the applicability of sophisticated and data-intensive models in flood damage assessments: (i) what is the added value, in terms of output quality and usefulness, attained by utilizing more detailed data and advanced methodologies? (ii) which are essential variables that play a key role in constraining the uncertainty bounds, making them worthy of investments in data collection?



The present paper aims at answering these questions, by leveraging the updating of the INSYDE model towards an use with the full treatment of input data uncertainty, involving the exploitation of detailed flood hazard and building inventories, here specifically developed and/or consulted for northern Italy, but with the potential for replication in any other contexts.

## 2 Materials and methods

### 2.1 From INSYDE to INSYDE 2.0

INSYDE is a synthetic, micro-scale, multi-variable flood damage model for the residential sector, released as an open-source R script, originally developed and validated for Italy, but extended also to Belgium (Dottori et al., 2016; Molinari et al., 2017; Scorzini et al., 2022). In INSYDE, the calculation of direct economic damages at the building scale relies on explicit, physically based mathematical equations describing flood damage mechanisms for each building component (and sub-components), as a function of more than 20 variables, including flood event (i.e., water depth, flow velocity, inundation

duration, sediment and pollution load) and building parameters (i.e., geometric and qualitative features (e.g.: footprint area, internal and external perimeter, building material, type and quality, etc.)), as well as prices for the reparation or replacement of the damaged items. For some building components, the damage mechanisms affected by greater uncertainties are modelled probabilistically by accounting for the probability of damage occurrence as a function of certain hazard intensity measures.

As stated in the Introduction, in case of missing information, the original model proposes deterministic default values for each input variable, calibrated on expert judgment and/or based on the analysis of large-scale local databases. Some of them, such as extensive parameters (e.g., internal area, external and internal perimeter of the building, etc.), are defined by default functional relationships calibrated on a typical configuration of a $100 \, \text{m}^2$ Italian house. According to Authors' experience, the implementation of INSYDE can then lead to biased results (due to the pairwise consideration of known and unknown

input data) or inaccurate estimations, especially when applied to large buildings, like apartment blocks, thus implying a scalability issue (Galliani et al., 2020).

Considering the sensitivity of damage estimates to individual input variables (albeit in varying degrees, not known a-priori), it is crucial to conduct a comprehensive analysis of the effects of missing information on model outcomes, by accounting also for both mutual and non-linear relationships among the variables. Such an approach can provide practical insights for

finding an efficient trade-off between model accuracy and efforts for input data retrieval (Di Bacco et al., 2023) and, then, a shift from the use of fixed deterministic values to suitable distributions of input variables could enhance users' awareness on damage estimation uncertainty.

By employing a step-wise procedure, the present study aims to address the aforementioned issues by proposing an updated version of INSYDE that will also enable the exploration of the two principal research questions outlined in the Introduction.

The methodological approach consisted of the main following phases (Figure 1):



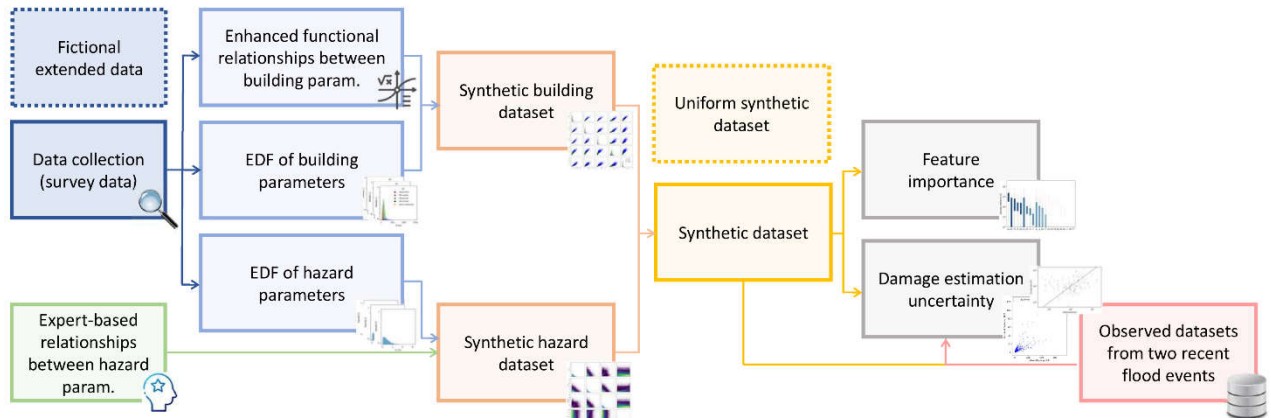

**Figure 1. Overview of the methodological approach. Dotted-line boxes represent an alternative dataset for the analysis, while maintaining the same methodological flow as depicted in the solid-line boxes.**

- Data collection (Section 2.2) to acquire relevant information on hazard and building features required by INSYDE;

- Development of INSYDE 2.0, incorporating a module for handling missing inputs in a probabilistic framework: this phase involved the generation of a synthetic dataset based on the collected empirical data combined with expert-based knowledge concerning relationships between hazard parameters (Sections 2.3 and 2.4);

- Assessment of model's sensitivities to missing input data: this phase included the analysis of the feature importance using the developed synthetic dataset, as well as the evaluation of the impact of individual or combined missing inputs on

uncertainty in damage estimation. This analysis has been conducted on a sampled building portfolio and on observed datasets for two recent flood events in Italy (Sections 2.5).

## 2.2 Procedure and data for updating INSYDE

The initial phase focused on establishing the foundation for a model capable of accurately capturing the hazard and building-specific details of the region of implementation. To achieve this, a "survey dataset" was developed as a basis for the

generation of empirical probability distributions (EDFs) for the variables at stake, which serve for sampling representative features of the populations of interest in case of unknown inputs for the application of INSYDE. In addition to traditional methods, such as deriving information from statistical data (Italian National Institute of Statistics, ISTAT), building inventories (OpenStreetMap, OSM) and field inspections, virtual surveys, entailing the analysis of building descriptions, floor layouts and photographs from real estate listings, were considered as useful means to collect information on micro-

scale building parameters influencing specific flood damage mechanisms (Scorzini et al., 2022) or to establish functional relationships among extensive parameters (e.g., internal and external perimeter as a function of footprint area) for different building typologies. The EDFs for describing the typical inundation phenomena in terms of water depth and flow velocity were instead derived from the analysis of the hazard maps produced for the 2021 update of the Flood Risk Management Plan of the Po River Basin District Authority (Autorità di Bacino del Fiume Po, 2022). Specifically, the selected maps consisted



in raster files obtained from 2D hydrodynamic modelling for high to low frequency (20- to 500-year return period) flood
scenarios in specific catchments of the district representing the distinctive inundation types occurring in rural or urban areas
as well as in flat or steeper regions within the Po River district. Given the limited availability of detailed information on
inundation duration and sediment load, expert knowledge was utilized to determine suitable distributions for these variables
that are able to capture the qualitative characteristics of typical flood events occurring in northern Italy. Due to the inherent

random nature of water pollution in flood events, a conservative assumption was made for the variable accounting for this
process, by assigning a 50% probability of having contaminated floodwater. Details on data statistics derived from the
analysis of ISTAT data, OSM building inventory, virtual surveys and flood-related data are available in the work by Huayra
Mena (2022).

**2.3 Major changes to the original model structure**

Based on the descriptive statistics obtained from the analysis of the data described in the previous section, the first
modifications to the original version of INSYDE concerned the resolution of the scalability issues associated with the use of
the functional relationships among extensive building parameters (Galliani et al., 2020). Following the strategy proposed for
the Belgian version of INSYDE (Scorzini et al., 2022), the housing unit (HU) has been chosen as the minimum calculation
item for multifamily buildings (i.e., condominiums). Typical ranges for the geometrical dimensions of individual HUs and

empirical formulas expressing the different extensive variables as a function of the building footprint area were determined
for each building type based on the samples gathered through the virtual survey in the investigated region. In addition, to
enhance and ease model's usability and to mitigate the impact of input data quality issues on the accuracy of damage
assessment, an algorithm has been implemented in INSYDE 2.0 to automatically split the building's footprint area into a
suitable number of HUs if the value introduced by the user significantly exceeds the maximum sizes observed in the

empirical samples. Such approach ensures that damage calculations are always performed at the HU scale, with the correct
tailored representative relationships among the variables at stake, even when input data are mistakenly provided at the whole
building scale, as it may happen in cases with limited data availability; only after this process, the resulting damages for each
HU are summed up to derive the overall estimate for the entire building.

**2.4 Towards an use of INSYDE 2.0 with an explicit treatment of input data uncertainty**

The probability distributions of the different input features representative of northern Italy for INSYDE 2.0 were generated
based on the available empirical hazard and building data, while also accounting for the intrinsic interdependence among the
variables (Tables 1 and 2). Specifically, the assumptions regarding the relationship between the building features relied upon
the empirical database and findings reported in Huayra Mena (2022), while a physically informed approach was adopted in
the case of the hazard variables, depending on the features characterizing both riverine (i.e., long-duration, low flow

velocity) and flash (i.e., rapid on-set, greater flow velocities and shallower water depth compared to other type) inundation
phenomena.



More in detail, probability distributions were first retrieved independently for the hazard variables based on detailed data when available (he, v) or upon expert-based assumptions derived from aggregated or approximated data (d, s, q), and used to sample sets of 250.000 elements; furthermore, the following functional dependencies were assumed to describe the

correlation among the features, based on the values sampled for he, d and v:

$$d^* = c_1 + c_2 \cdot \sqrt{he} \cdot N(\mu = 1, \ \sigma = 0.2)$$

$$v^* = c_3 - d/\max(d) \cdot N(\mu = 1, \ \sigma = c_3 - d/\max(d))$$

$$s^* = c_4 + c_5 \cdot \sqrt{v} \cdot N(\mu = 1, \qquad \sigma = 0.2)$$

with N being a random number from a normal distribution with mean $\mu$ and standard deviation $\sigma$; q was instead assumed

independent from the other hazard features.

The resulting d*, v* and s* are auxiliary datasets describing the correlation among the hazard variables, but in general they do not follow the probability distributions retrieved independently for d, v and s; to obtain the correct distributions without losing information on the interdependence among the variables, the values of d*, v* and s* were then replaced with the correspondent percentiles from the datasets of d, v and s.


**Table 1. Hazard features considered in INSYDE 2.0 and assumed probability distributions for the case of northern Italy.**

| Variable | Description | Distribution |
|---|---|---|
| he | Water depth [m] | Weibull minimum (shape=1.25, scale=1); if h < 0.01 resampled from Uniform{0.01, 0.03} |
| d | Inundation duration [hours] | Weibull minimum (shape=1.25, scale=36); if d < 1 resampled from Uniform{1, 2} |
| q | Presence of pollutants [yes (1) / no (0)] | P(q=0) = 0.5, P(q=1) = 0.5 |
| v | Velocity [m/s] | Weibull minimum (shape=1.15, scale=0.35); if v < 0.05 resampled from Uniform{0.05, 0.1} |
| s | Sediment load [-] | Uniform{0.05, 0.2} |

**Table 2. Building features considered in INSYDE 2.0 and assumed probability distributions for the case of northern Italy.**

| Variable | Description | Distributions |
|---|---|---|
| BT | Building type [-] | *ECDF based on ISTAT data*<br>P(BT=1 (Detached))=0.54; P(BT=2 (Semi-detached))=0.10; P(BT=3 (Apartment))=0.13; P(BT=4 (Attached corner))=0.10; P(BT=5 (Attached center))=0.13 |
| FA | Footprint area [m$^2$] | *ECDF based on OSM data*<br>Truncated normal ($\mu = 160$, $\sigma = 60$, min=50, max=320) if BT=1<br>Truncated normal ($\mu = 110$, $\sigma = 20$, min=50, max=160) if BT=2<br>Truncated normal ($\mu = 95$, $\sigma = 20$, min=60, max=160) if BT=3<br>Truncated normal ($\mu = 85$, $\sigma = 15$, min=45, max=140) if BT=4<br>Truncated normal ($\mu = 85$, $\sigma = 15$, min=45, max=120) if BT=5 |
| IA | Internal area [m$^2$] | $0.9 \cdot FA$ |
| BA | Basement area [m$^2$] | $0.5 \cdot FA \cdot Normal(\mu = 1, \sigma = 0.2)$ |
| EP | External perimeter [m] | *Empirical relationships identified from the analysis of OSM data*<br>$4.1 \cdot \sqrt{FA} \cdot Normal(\mu = 1, \sigma = 0.2)$ if BT=1 |

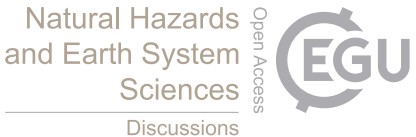
| | | |
|---|---|---|
| | | $3 \cdot \sqrt{FA} \cdot$ Normal($\mu$= 1, $\sigma$=0.2) if BT=2 or BT=4 |
| | | $-6.9729 + 0.2885 \cdot$ FA $\cdot$ Normal($\mu$= 1, $\sigma$=0.2) if BT=3 |
| | | $2 \cdot \sqrt{FA} \cdot$ Normal($\mu$= 1, $\sigma$=0.2) if BT=5 |
| IP | Internal perimeter [m] | *Empirical relationships identified from the analysis of the data from the virtual surveys*<br>$20.151 + 0.6254 \cdot$ FA $\cdot$ Normal($\mu$= 1, $\sigma$=0.2) if BT=1<br>$20.119 + 0.6105 \cdot$ FA $\cdot$ Normal($\mu$= 1, $\sigma$=0.2) if BT=2<br>$20.336 + 0.6576 \cdot$ FA $\cdot$ Normal($\mu$= 1, $\sigma$=0.2) if BT=3<br>$9.709 + 0.6902 \cdot$ FA $\cdot$ Normal($\mu$= 1, $\sigma$=0.2) if BT=4<br>$16.801 + 0.559 \cdot$ FA $\cdot$ Normal($\mu$= 1, $\sigma$=0.2) if BT=5 |
| BP | Basement perimeter [m] | *Empirical relationships identified from the analysis of the data from the virtual surveys*<br>$4.2 \cdot \sqrt{FA} \cdot$ Normal($\mu$= 1, $\sigma$=0.2) |
| NF | Number of floors [-] | *ECDF based on ISTAT data*<br>P(NF=1) = 0.09, P(NF=2) = 0.56; P(NF=3) = 0.25; P(NF>3) = 0.10 |
| IH | Interfloor height [m] | *ECDF based on survey data*<br>Virtual Survey ECDF + Truncated normal ($\mu$= 0, $\sigma$=0.5, min=-0.15, max=0.15) |
| BH | Basement height [m] | *ECDF based on survey data*<br>Skewed normal (skewness= -4, $\mu$= 3, $\sigma$=0.25) |
| GL | Ground floor level [m] | *ECDF based on survey data*<br>Normal($\mu$= 0.1, $\sigma$=0.09) |
| BL | Basement level [m] | -GL-BH-0.3 |
| BS | Building structure [-] | *ECDF based on ISTAT data*<br>P(BS=1 (Reinforced concrete)) = 0.33; P(BS=2 (Masonry)) = 0.67 |
| FL | Finishing level (i.e. building quality) [-] | *ECDF based on survey data*<br>P(FL=0.8 (Low)) = 0.05, P(FL=1 (Medium)) = 0.42; P(FL=1.2 (High)) = 0.53 |
| LM | Level of maintenance [-] | *ECDF based on ISTAT data*<br>P(LM=0.9 (Low)) = 0.13, P(LM=1 (Medium)) = 0.47; P(LM=1.1 (High)) = 0.40 |
| YY | Year of construction [-] | *ECDF based on ISTAT data* |
| PD | Heating system distribution [-] | *ECDF based on the analysis of grey-literature and survey data*<br>if YY $\geq$ 1990: P(PD=1 (Centralized)) = 0.11, P(PD=2 (Distributed)) = 0.89<br>if YY < 1990: P(PD=1) = 0.6, P(PD=2) = 0.4 |
| PT | Heating system type [-] | *ECDF based on the analysis of grey-literature and survey data*<br>if YY > 2000 & FL > 1: P(PT=1 (Radiator)) = 0.2, P(PT=2 (Pavement)) = 0.8<br>else P(PT=1) = 0.8, P(PT=2) = 0.2 |
| BE | Basement exists [-] | *ECDF based on survey data*<br>P(BE=0 (No)) = 0.2, P(BE=1 (Yes)) = 0.8 |

## 2.5 Model's sensitivities to missing input data

### 2.5.1 Analysis of the feature importance

In the new framework for missing data handling, the generated synthetic dataset has been exploited in a feature importance exercise aimed at a quantitative assessment of the sensitivity of damage calculations to the absence of information on certain input variables, in order to identify key features deserving attention in data collection. This analysis, based on a probabilistic test performed on a complete subset of 5000 hypothetically flooded buildings, involved the following steps: first, INSYDE 2.0 is used to calculate damage on this complete dataset, where all input values are assumed to be available, and the resulting estimate is taken as a reference point; next, the values of one input variable are removed at a time from the dataset, and the corresponding missing values are sampled from the generated synthetic dataset. This process is repeated for each variable





and, each time, damage is recalculated; the difference in damage with respect to the reference value is finally recorded and then the variance induced by each feature on model outcome can be determined.

**2.5.2 Analysis of damage estimation uncertainty**

In addition to assessing the possible contribution of unknown single input features to damage estimation uncertainty, a further analysis was carried out to evaluate the impact of the combined absence of multiple input variables on the variability of damage estimations.

2.5.2.1   Analysis on the synthetic dataset

A first application of INSYDE 2.0 was conducted on the complete synthetic portfolio of 5000 buildings mentioned in the previous section, this time altered to account for the presence of multiple unknown data within the tested sample. The reduction in the dataset's level of completeness was achieved by assuming different percentages of missing data for each feature, which were assigned based on their typical availability or ease of retrieval, as experienced by the Authors in the Italian context. Except for he and FA, which were considered as the minimum known variables for a damage assessment, the

missing values were placed randomly, as follows: 10% for variables of easy retrieval, either due to their availability at the meso-scale (e.g., census block scale) or to their low variability (BT, IH, NF, BS, LM, FL, YY) and 20% for other building features that require specific surveys for correct characterization (EP, BE and related variables, BH, BA, BP); for GL, which is generally not available in databases, but potentially appraisable through (virtual) surveys, this percentage was increased to 50%, while 95% was assumed for the building features that are hardly ever known (or only after internal surveys), such as IP

and PD. For the hazard variables, the percentages were assumed to be 10, 20, 50 and 80%, respectively for v, d, s and q, taking into account the increasing modelling costs from a simple 2D steady hydrodynamic simulation to a more complex unsteady run with the inclusion of sediment transport modelling; the very specific and detailed data requirements regarding the presence and propagation of pollutants instead explain the higher value assumed for q. For each tested object, 1000 complete replicates were generated by filling missing input data with values sampled from the developed synthetic dataset

and the corresponding average damage and standard deviation were calculated.

2.5.2.2   Analysis on field data from recent flood events

A similar analysis was also carried out considering real-world, field databases compiled for two flood events that occurred in the Po River valley: the 2002 Adda flood in Lodi and the 2010 Bacchiglione flood in Caldogno, both of which have been described in previous applications of INSYDE (Dottori et al., 2016; Amadio et al., 2019; Molinari et al., 2020). Table 3

provides a concise overview of the available datasets, by specifically highlighting the unknown variables for INSYDE 2.0 in the two case studies. As typical in large-scale flood damage assessments, the missing data mainly concerned the ultra-detailed characteristics of the dwellings, while only approximate information on inundation duration was available from the reports of the events, which provided a rough indication of 24 hours on average for both cases. To ensure a reasonable level





of uncertainty while considering the available information on inundation duration, the empirical distribution for this variable
was modified with respect to the one in Table 1, by sampling d values from an assigned truncated normal distribution
centered at 24 hours and spanning between 16 and 48 hours. As in the previous case, the approach entailed calculating
damage over 1000 complete replicates for each affected building and registering the corresponding damage statistics.

Furthermore, considering the availability of observed losses for the two case studies, we also investigated the impact of
missing inputs on the results of classical validation exercises, raising questions on general  interpretation of their results
when performed for simple (e.g., univariable) or complex models without a proper treatment of uncertainties (Molinari et al.,
2019, 2020). In this context, since its formulation, INSYDE has undergone continuous updates and validation, with reported
superior performance when compared to other tested damage models (Dottori et al., 2016; Amadio et al., 2019; Molinari et
al., 2020). Although these previous studies consistently demonstrated INSYDE's capacity to provide accurate damage
estimations, the reliance on fixed default values for missing input data limited the quantitative assessment of the uncertainty
associated with validation outcomes.

**Table 3. Unknown input features for INSYDE in the considered validation case studies.**

| Case study | Unknown input features in the dataset |
|---|---|
| Lodi | d*, s, IP, IH, BH, PD, PT, BA, BP (for all buildings (271) in the dataset) <br> GL and NF (partial availability - known, respectively, for 47 and 265 buildings) |
| Caldogno | d*, s, q, IP, IH, BH, GL, PD, PT, LM, BA, BP, HB (for all buildings (294) in the dataset) |

## 3 Results and discussion

### 3.1 Generation of the synthetic dataset

The pair plot shown in Figure 2 enhances the visualization of the pairwise relationships assumed in this study among the
flood hazard variables, water depth (he), flow velocity (v), inundation duration (d) and sediment load (s). This graphical tool
employs a scatter plot to illustrate the relationship between each pair of variables in the dataset, while the diagonal axis
indicates the distribution of each variable. From the patterns represented in Figure 2 it is evident the physically informed
approach adopted for hypothesizing the relationships among the variables: for instance, a positive relationship between he
and d, as well as between v and s, with the latter explained by the tendency of flash floods to carry greater amounts of debris;
similarly, d and v were considered to be negatively correlated, in consideration of the short duration typically associated with
flash floods.

An analogous pair plot for the extensive building variables is presented in Figure 3, which illustrates the functional
relationships (Table 2) identified from the analysis of the empirical survey dataset (Huayra Mena, 2022).


**Figure 2. Empirical pairwise relationships assumed for the generation of the distributions representative of northern Italy: hazard parameters (water depth (he), flow velocity (v), inundation duration (d) and sediment load (s)).**

For the sake of clarity, it should be noted that the distributions for the "Apartment" category are represented in Figure 3 at the building block scale, having assumed a number of housing units (nHU ≥1) generated from a Weibull distribution with shape and scale parameters equal to 2 and 4, respectively. The pair plots reported in the Supplementary material (Figures S1 and S2), while preserving the nature of the identified functional relationships among the variables, but spanning over wider





ranges of them, were also generated to support a more comprehensive investigation of the feature importance within INSYDE 2.0, regardless of the specific characteristics of the sample representative of northern Italy.



**Figure 3. Empirical pairwise relationships assumed for the generation of the distributions representative of northern Italy: extensive building parameters (footprint (FA) and basement (BA)area; external (EP), internal (IP) and basement perimeter (BP)).**

### 3.2 Model's sensitivities to missing input data

### 3.2.1 Analysis of the feature importance

This section reports on uncertainty in damage calculations resulting from the potential lack of knowledge on certain input data in INSYDE 2.0. In detail, Figure 4 summarizes the results of the feature importance analysis by showing the difference





in computed damage when applying the model to a reference complete synthetic set of 5000 buildings and to their replicas obtained by replacing the values of one input variable at a time with a sampling from the dataset generated for the northern Italy case (upper panel) or from the extended uniform synthetic dataset reported in the Supplementary material (lower panel).

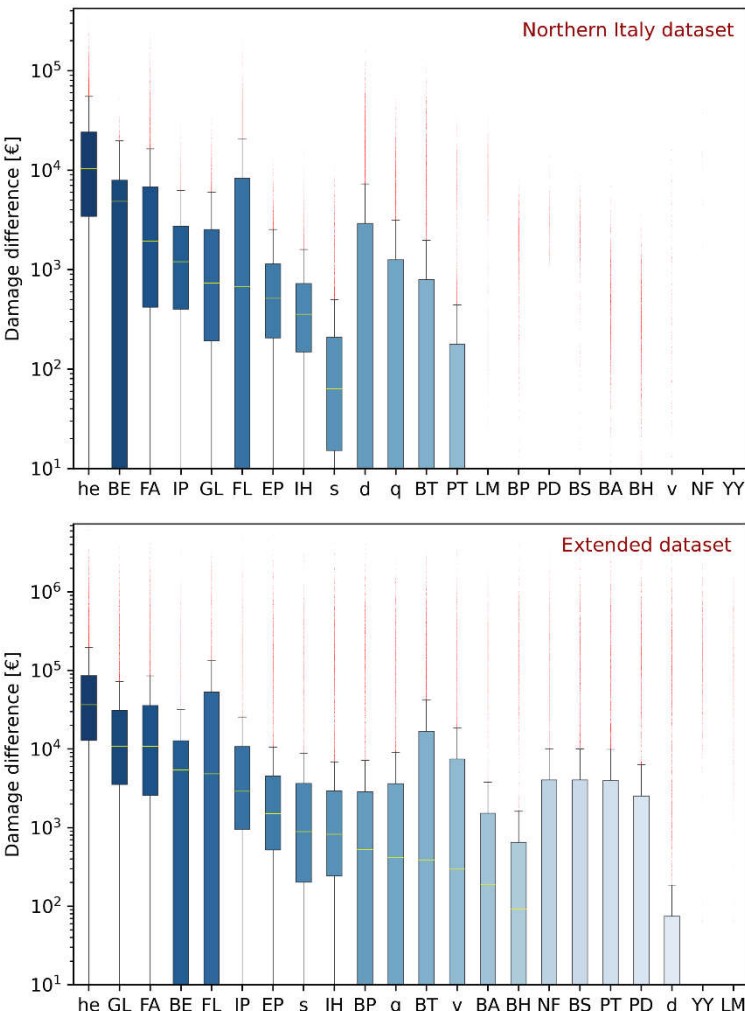

**Figure 4.** Feature importance in INSYDE 2.0: (upper panel) test with sampling from the synthetic dataset developed for northern Italy; (lower panel) test with sampling from the uniform synthetic dataset (ref. to Figures S1-S2). The variables are ranked according to the median value (in yellow) of the estimated damage difference with respect to the reference damage calculated on a complete dataset.

Consistently with the literature (Kelman and Spence, 2004; Schröter et al., 2014; Dottori et al., 2016; Amadio et al., 2019; Scorzini et al., 2022), Figure 4 confirms the importance for flood damage modelling of relying on accurate input data for water depth, even though damage differences associated to it are found to be, on average, around only 10.000 Euro (upper panel), due to the intrinsic limited variability assumed for this variable in the generation of the empirical distributions for the context of northern Italy (Figure 2). Albeit with a comparatively lower influence, sediment load, inundation duration and the





indicator for the presence of pollutants can be ranked as other important hazard input features, with the latter two inducing more variability in the results, as a consequence of some damage mechanisms activated in INSYDE on the basis of thresholds on d or q (Dottori et al., 2016; for clarity, an example of such damage mechanism is reported in the Supplement). The riverine inundation characteristics, typical of the examined context (Figure 2) and insufficient to cause structural damages (Clausen and Clark, 1990), also explain why a lack of input data on flow velocity does not induce any tangible effect on damage estimation. A different pattern is instead visible in the lower panel of Figure 4, obtained from a sampling based on the extended synthetic dataset (Figures S1 and S2), featuring larger ranges of values for the tested input variables and thus providing more general insights on model sensitivity to input data availability (regardless of the specific local characteristics for the context of model customization). In this case, apart from the greater differences observed in absolute terms, the figure indicates that velocity has a far more relevant impact than inundation duration on damage estimation uncertainty when dealing with long-lasting flood events (as represented in the extended synthetic dataset, Figure S1), exceeding the duration threshold assumed for certain damage mechanisms (Dottori et al., 2016; please refer to the code of INSYDE 2.0 for details).

Regarding building features, the upper panel of Figure 4 reveals the significant and obvious influence of the extensive variables (FA, IP, EP), of the binary variable BE for the presence of the basement (which masks the importance of the basement-related variables, BA, BP and BH), and building's elevation with respect to the ground level (GL). Finishing level (FL) causes relevant variability on model outcomes, with an observed median damage difference of about 670 € for the northern Italy data, while a detailed knowledge on variables such as level of maintenance (LM), building structure (BS) and heating distribution (PD) type, and even more the number of floors (NF) and the year of construction (YY), appear to provide an overall negligible impact on damage estimation uncertainty. Again, such results are dependent on the specific datasets used for sampling missing values and, therefore, for a more general overview on the ranking of the feature importance in INSYDE 2.0, it is possible to refer to the lower panel of Figure 4, which illustrates how some variables (such as NF, BS, PD and PT) gain increasing importance when hazard parameters are set to (larger) values capable of activating damage mechanisms to more building components. These findings then demonstrate how the importance of specific input parameters can vary depending on the characteristics of the study region, thus highlighting the cruciality of relying on regionally representative hazard and building datasets for an enhanced and efficient flood damage modelling.

### 3.2.2 Analysis of damage estimation uncertainty

#### 3.2.2.1 Analysis on the synthetic dataset

Figure 5 reports the results of the analysis aimed at evaluating the performance of INSYDE 2.0 when the absence of multiple inputs is considered. In detail, the figure shows the mean damage and standard deviation calculated for each of the 5000 modified (i.e., with multiple missing inputs) items over their 1000 complete replicates generated by populating the missing information with values sampled from the defined synthetic dataset for northern Italy. Interestingly, the figure shows that, for





all the building typologies, the results tend to lie on two different trend lines corresponding to higher or lower damage variability. A closer inspection of the results reveals that these distinct patterns are not necessarily related to the quantity of missing variables, but rather to their role in the damage mechanisms implemented in INSYDE. Indeed, for certain building components, the estimated damage depends on the occurrence of certain conditions on multiple variables; in such cases,

300     when more than one of these conditions are met, the maximum resulting damage is assumed to hold, since the most unfavorable state is thought to dominate the damage mechanism, irrespective of the others (Dottori et al., 2016). An example of this situation is represented by the components related to the interior or exterior plaster (details in the Supplementary material as well as in the code), for which damage occurrence is supposed to depend on inundation duration and flow velocity, with a varying degree expressed by the corresponding fragility functions, as well as on water quality (q) and level

305     of maintenance (LM) of the building, with a 100% probability of damage occurrence in case of contaminated water (q=1) or average/poor level of maintenance (LM≤1); in particular, the conditions applied to the last two variables eliminate any potential estimation uncertainties arising from missing data on other parameters involved in the damage mechanism. A similar uncertainty-limiting behavior is also distinctive of damage to pavement components, which theoretically depend on different input features, but only when finishing level (FL) is set to certain values (FL>1).

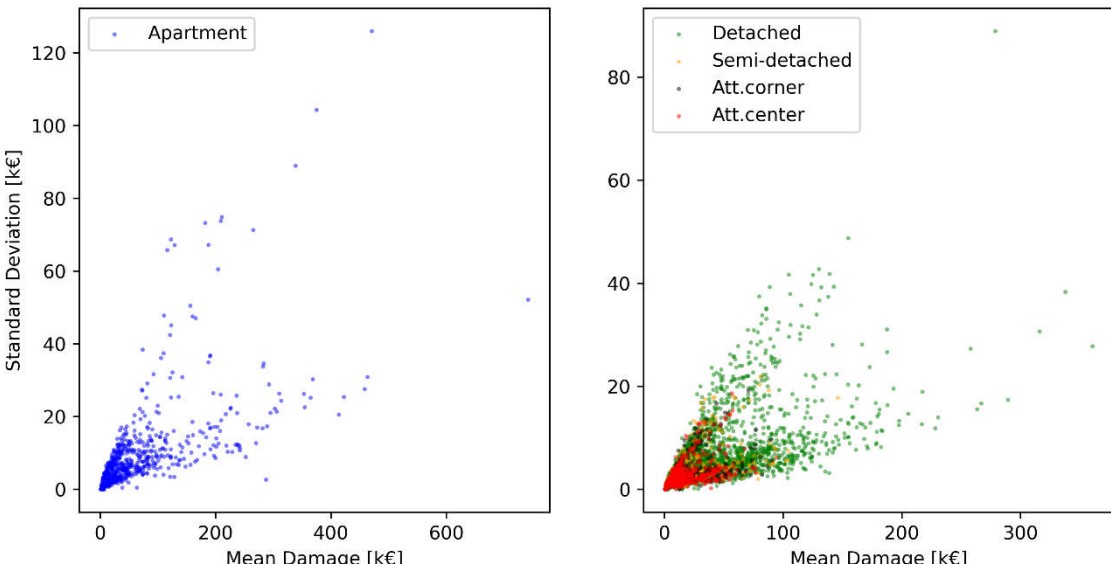

310

**Figure 5. Damage estimation variability observed on the altered (i.e., with randomly generated multiple missing inputs) synthetic dataset representative of northern Italy.**

3.2.2.2    Analysis on field data from recent flood events

Similar trend patterns to those presented in Figure 5 are also evident in Figure 6, which displays the results obtained by

315     replicating the data filling procedure to the empirical datasets for the flood events in Lodi and Caldogno, both of which originally characterized by the presence of some unknown input features (Table 3). The minor differences visible between the two case studies (Figure 6) are again a consequence of the type of missing variables within each dataset.

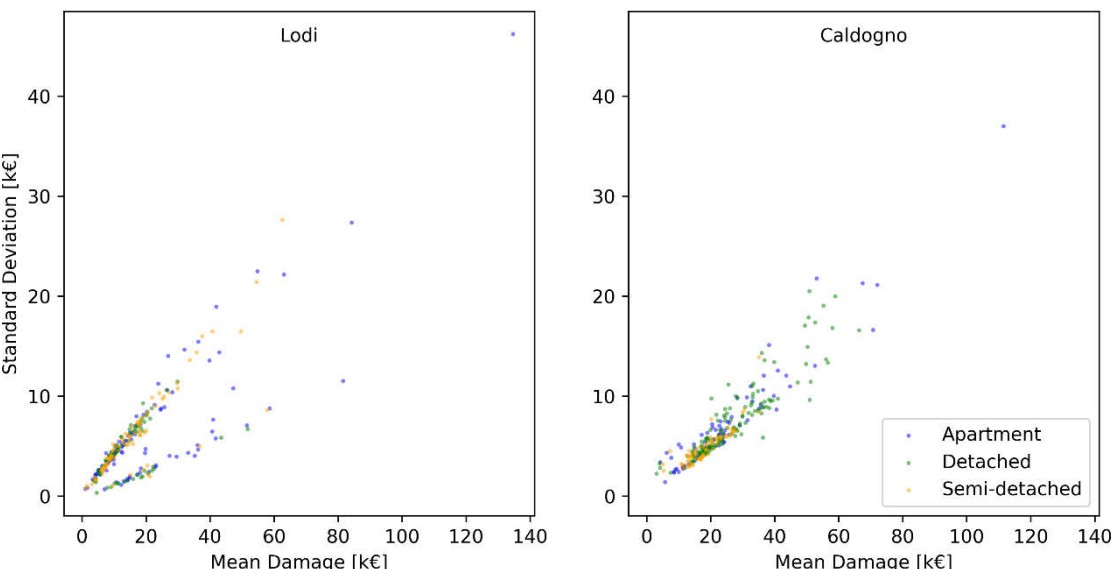

**Figure 6. Damage estimation variability observed on the empirical datasets for the case studies of Lodi and Caldogno.**

Specifically, the points lying on the lower variability trend line for the Lodi case are representative of those buildings with available information on GL, which significantly reduces damage estimation uncertainty. If excluding these data, Lodi generally exhibits slightly larger standard deviations for the same calculated mean damage in Caldogno. Such difference can be explained by considering the input data availability in the two cases for certain key variables (q and LM) which can act as limiting or amplifying factors of damage estimation variability. In detail, complete information on these key variables is only available for the Lodi dataset, with just a restricted number of buildings exhibiting the mentioned "uncertainty limiting values" q=1 and LM≤1 (respectively in ~6% and ~15% of the elements in the dataset).

The two cases studies were also considered to highlight the value of the proposed approach in interpreting the results of model validation, particularly important for a complex multi-variable damage model like INSYDE. The outcomes of the test are summarized in Table 4, which compares total observed damages against damage statistics obtained by applying INSYDE 2.0 over 1000 replicates of each affected item in the two building portfolios containing missing input features.

These findings are complemented by Figure 7 which offers a visual representation of the detected differences between estimations and observations at the individual building scale. Table 4 illustrates a general convergence between observed and estimated damages, particularly around the 75[th] percentile, where the calculated losses align with the reported values. The median estimates exhibit a satisfactory level of agreement with the observed losses, which is consistent with typical outcomes observed in validation exercises for models demonstrating overall good performances (e.g., Amadio et al. 2019; Molinari et al., 2020). It should be noted that the model tends to overestimate lower entity damages across all building types (Figure 7), but this discrepancy, rather than being a consequence of any model-related issue, can be primarily attributed to



the limitations in the representativeness of claim data, particularly for minor losses, as documented in the literature (Merz et

al., 2008; Molinari et al., 2020; Pinelli et al., 2020).



**Figure 7. Results of the probabilistic validation of INSYDE 2.0 for the case study of Lodi (left) and Caldogno (right). Median computed damage (dot) and corresponding interquartile range (line) are plotted for each building against observed damage (expressed in 2021 Euro).**






**Table 4. Statistics of total calculated damage with INSYDE 2.0 versus reported damages for the considered case studies.**

| Case study | Calculated damage [M€ 2021] | | | | | Observed damage [M€ 2021] |
|---|---|---|---|---|---|---|
| | 5th percentile | 25th percentile | median | 75th percentile | 95th percentile | |
| Lodi | 3.13 | 3.68 | 4.18 | 5.26 | 8.06 | 5.05 |
| Caldogno | 3.46 | 6.44 | 7.53 | 8.54 | 10.34 | 8.35 |

While confirming the performance of INSYDE 2.0 in accurately depicting the overall damage figures of the two events
(Dottori et al., 2016; Amadio et al., 2019; Molinari et al., 2020), the results of this analysis also emphasize the advantages of
incorporating the treatment of input data uncertainty when presenting the outcomes of a model validation. Indeed, this
approach enhances the robustness and reliability of the model, by providing a clear indication of the uncertainty bounds of
the estimations, thus effectively mitigating the risk of conveying a false perception of certainty, which instead may be
encountered with simple deterministic approaches or even with more sophisticated models when used in combination with
oversimplified assumptions (Merz et al., 2005; Pappenberger and Beven, 2006).

## 4 Conclusions

Accurately assessing flood risk is crucial for mitigating the potentially devastating effects of flooding. However, the
complexity of the systems involved, and the significant amount of data required, make flood damage estimation a
challenging task, susceptible to uncertainties from input data, model structure and assumptions. Achieving a trade-off
between outcome reliability (with a quantitative characterization of uncertainty) and estimation efforts (in terms of time and
financial resources for both data retrieval and modelling) is essential for efficient and comprehensive risk assessments,
enabling optimal decision-making (Apel et al., 2008; Merz et al., 2015; Sieg et al., 2023). To strike this balance, it is
important to examine the possible added value of utilizing more detailed data and advanced methodologies, as well as
identifying critical variables that reduce damage estimation uncertainty, justifying investments in data collection.
In this context, the present study aimed at addressing these issues through the development of an updated version of a multi-
variable flood damage model, INSYDE, which estimates direct economic damages at the building scale as a function of
several flood event and building features. In consideration of the amount and detail of required input variables, the process of
data retrieval and preparation for a multi-variable model, like INSYDE, can be resource-intensive and incomplete inputs can
significantly impact on the variability of calculated damages. The proposed updated version of INSYDE incorporates a
probabilistic module for filling missing input data, offering a transparent information on uncertainties arising from limited
knowledge on damage explicative variables. This approach ensures more reliable and robust assessments, reducing the risk
of conveying a false perception of certainty that can occur when using univariable or simple deterministic approaches, even
when interpreting the results of model validation exercises (Merz et al., 2005; Pappenberger and Beven, 2006; Amadio et al.,



2019; Molinari et al., 2019, 2020). In this context, the present study demonstrates the value of generating comprehensive
synthetic datasets of flood hazard and building features that can be leveraged to identify key variables worthy of specific
investments in data retrieval. Indeed, the development and use of synthetic datasets, combined with uncertainty analysis on
model outcomes, can help in bridging the data gaps and addressing the challenges associated with the availability and
completeness of input variables.

Obtained results also indicate that, in addition to the standard hazard variables, an accurate description of the building
features is essential to derive reliable estimations of flood damage (Schröter et al., 2018; Molinari et al., 2020; Taramelli et
al., 2022). While data retrieval on large-scale for some of the vulnerability variables can be costly (Ruggieri et al., 2021), the
use of the proposed probabilistic missing data filling procedure, based on representative datasets of the local building stock,
can be employed as an option. This can not only help to solve the problem of insufficient knowledge about certain input
features (Pinelli et al., 2020; Gómez Zapata et al., 2022), but also to provide decision makers with a better understanding of
the uncertainty associated with the estimations (Razavi et al., 2021). Moreover, the results of the feature importance analysis
conducted in this study highlights the significance of relying on representative datasets capturing the characteristics of the
investigated area for a proper identification of the key variables to be considered when modelling flood damage.

The process for developing these specific datasets, here exemplified for northern Italy (Po River district), but theoretically
replicable in any other region/country, mainly involves a combination of traditional methods for data collection, such as
desk-based analysis of statistical databases as well as virtual surveys for creating building portfolios; even though such tasks
can be time-consuming, especially in consideration of the significant regional spatial variability of the building stock, it is
worth noting that emerging technologies, such as remote sensing and automatic image reconnaissance (Velez et al., 2022),
can potentially enhance the process in the future, with a more efficient and accurate exposure and vulnerability modelling.

In conclusion, this study demonstrates the significant added value of adopting a probabilistic approach with the explicit
treatment of input data uncertainties, thus providing insights for more informed risk assessments, while ensuring efficient
data collection procedures. Overall, it also emphasize the ongoing importance of refining data collection and modelling
approaches, since a comprehensive and reliable characterization of inundation phenomena and impacted assets is crucial for
enhancing confidence in the outcomes of damage assessment processes.

**Acknowledgements**

The Authors gratefully acknowledged Grecia Geraldine Huayra Mena for contributing to the first phase of the present study
during her M.Sc. and the Po River District Authority for supplying the data required for the investigation of hazard features
in Northern Italy.

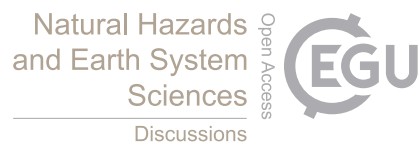

## Code availability

The code of INSYDE 2.0 is available at the following link https://drive.google.com/file/d/16EpvE-xkmh-
ivaSxzfq540HuaP_sXrtc/view?usp=sharing (during the review process) and it will be stored in Mendeley Data upon
acceptance.

## Authors contribution

Conceptualization: M.D.B., D.M. and A.R.S.; Data curation: M.D.B. and A.R.S.; Formal analysis: M.D.B. and A.R.S.;
Investigation of results: M.D.B., D.M. and A.R.S.; Software: M.D.B.; Visualization: M.D.B. and A.R.S.; Writing – original
draft: A.R.S.; Writing – final draft: M.D.B., D.M. and A.R.S.

## Competing interests

The authors declare that they have no conflict of interest.

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
