# Peer review of "The value of ultra-detailed survey data for an improved flood damage modelling with explicit input data uncertainty treatment: INSYDE 2.0"

_Natural Hazards and Earth System Sciences, 2023_

## Author Comment (AC2)

We would like to thank the Reviewer for his/her interest in our work and for carefully reading our manuscript; we greatly appreciate the insightful comments as they contribute to increase the manuscript robustness and, in general, to improve its quality. In the following, we provide a point-by-point reply to the general and specific comments raised.
* * *
**REVIEWER 2**

The paper introduces INSYDE 2.0, a flood damage modeling tool that integrates ultra-detailed survey and desk-based data to enhance the reliability and informativeness of flood damage predictions. By incorporating a probabilistic module, it addresses missing input data and provides transparent information about uncertainties arising from limited knowledge of damage explicative variables. This integration significantly improves the reliability and robustness of flood damage assessments. While the extensive use of ultra-detailed data contributes to the model's reliability and the importance of accurate damage estimation, the logical coherence and readability of the manuscript still require significant improvement. Therefore, a major revision is necessary to strengthen the manuscript for potential publication.

**Major comments**:

**R2.C1.** One of my main comments is regarding the dataset used in this study. The manuscript introduces various datasets, such as synthetic datasets, observed datasets, survey datasets, auxiliary datasets, specific datasets, and empirical datasets. However, it is unclear what each dataset represents and how they are related. Are these datasets intended to establish the functional relationships between building parameters, serve as reference data, or facilitate model validation? The lack of clarity regarding the nature and relationships of these datasets, as well as their correspondence with the terms used in Figure 1, significantly hampers readability. I think it's good to provide a bit more information on the construction of all these datasets. Particularly since this manuscript focuses on uncertainty analysis.
**Reply**: We agree with the Reviewer that the original description of the various datasets involved in our study may lead to confusion. In the revised version, we will provide a more precise and clear explanation of each dataset's purpose and how they interrelate. This clarification will be particularly emphasized in the revision of Figure 1, ensuring a better understanding of the roles these datasets play in establishing functional relationships between building parameters, serving as reference data, and facilitating model analysis and application. This adjustment will aim to enhance the overall readability and coherence of the manuscript.

**R2.C2.** The description of building damage in the paper is not very intuitive, and the location distribution information of buildings is hardly mentioned, so it is difficult for readers to apply INSYDE 2.0 to their examples. In my opinion this question - how do we sample complex and spatially distributed variables in a meaningful way? - is one of the key research questions that the uncertainty and sensitive analysis community will need to work on if we want to move on to the next stage of applying this type of technique to complex models. I am not suggesting the authors should solve this issue, but I think they should point out this is a very important and critical step and a big area for future research and development.
**Reply**: In the revised manuscript, we will enhance the description of the study area, including the incorporation of a map to provide a clearer understanding of the geographical context. We acknowledge that addressing the question of how to effectively sample input variables for damage models is a key challenge and an area for future research and development within the community. In our paper we have proposed an attempt for solving this issue (with a framework that can be replicated also in other areas), by deriving representative distributions from multi-source data, which allow to sample unknown input features at the local scale.

**R2.C3.** In Section 2.2, the meaning of "specific catchments of the district representing the distinctive inundation types occurring in rural or urban areas as well as in flat or steeper regions within the Po River district" is unclear. It would be beneficial to include a spatial distribution map or a dem illustrating the Po River district to provide a better understanding of the study area.
**Reply**: A map of the Po River District will be incorporated in the revised version of the manuscript to provide a visual representation of the study area.

**R2.C4.** While it is described that nearly 5000 hypothetically flooded buildings are analyzed, the manuscript does little to explain or describe the building characteristics involved. Numerous contributing factors affect

economic loss. Given the primary objective of this manuscript is to quantify loss, it would be necessary for the authors to describe the 5000 hypothetically flooded buildings in more detail.

**Reply**: In the revised manuscript, we will make clear that the 5000 hypothetically flooded buildings are sampled from the empirical distributions presented in the original Table 2, which already comprehensively describes the housing stock in the Po River District.

**R2.C5**. In lines 298-301, I guess the authors mean that high variability in certain parameters leads to significant deviations in loss estimation. If so, this issue represents a substantial portion of the cases and should be addressed. The authors may consider discussing potential measures to reduce such errors.

**Reply**: In the specified lines, the text indeed addressed the fact that high variability in certain parameters can lead to significant deviations in loss estimation. This issue was thoroughly explored in the section discussing the feature importance, where it was acknowledged that certain variables carry more importance than others due to their role played on damage mechanisms. Additionally, another aspect is also discussed: it points out that variations in input parameters may only result in damage variations if certain conditions on other parameters are met.

Regarding the suggestion to identify measures to reduce "errors," it is important to note that these are not errors, but uncertainties. Instead of aiming to reduce uncertainty, the focus of the study is on providing explicit information about it and highlighting the importance of the features that contribute to this uncertainty and this perspective will be maintained in the revised version of the manuscript.

**Minor comments**:

**R2.C6**. Language and Sentence Structure: Please carefully review the manuscript for sentence clarity and readability. Some long sentences, such as lines 301-303, are difficult to follow. Consider breaking them into two sentences or rewriting them for better readability.

**Reply**: In revising the manuscript, we will enhance the overall readability of the text, by paying particular attention to longer sentences to make them clearer, potentially by breaking them into two sentences or rephrasing them for better readability.

**R2.C7**. Line 154: Typo - "250.000" should be corrected.

**Reply**: This will be fixed in the revised version of the manuscript (250'000).

**R2.C8**. Line 156-158: Please describe the meaning of values "$c_1$-$c_5$."

**Reply**: They are constant values introduced in the expert-based approach to obtain the desired functional relationships among the variables. We acknowledge that this may not have been explicitly mentioned in the original manuscript and we will therefore clarify it in the revised version.

**R2.C9**. Line 163-164: Clarify the meaning of "to obtain the correct distributions without losing information on the interdependence among the variables, the values of d*, v*, and s* were then replaced with the correspondent percentiles from the datasets of d, v, and s."

**Reply**: This sentence will be improved for clarity in the revised version of the manuscript as follows:

"[…] More in detail, probability distributions were first retrieved independently for the hazard variables based on detailed data when available (he, v) or upon expert-based assumptions derived from aggregated or approximated data (d, s, q), and used to sample sets of 250.000 elements; furthermore, the following functional dependencies were assumed to describe the correlation among the features, based on the values sampled for he, d and v:

$$d^* = c_1 + c_2 \cdot \sqrt{he} \cdot N(\mu = 1, \ \sigma = 0.2)$$

$$v^* = c_3 - d/\max(d) \cdot N(\mu = 1, \ \sigma = c_3 - d/\max(d))$$

$$s^* = c_4 + c_5 \cdot \sqrt{v} \cdot N(\mu = 1, \qquad \sigma = 0.2)$$

with N being a random number from a normal distribution with mean $\mu$ and standard deviation $\sigma$, while the coefficients $c_i$ are constant values introduced in the expert-based approach to obtain the desired functional relationships among the variables. q was instead assumed independent from the other hazard features.

Although the resulting d*, v* and s* account for the correlation among the hazard variables, they do not follow the probability distributions retrieved independently for the variables d, v and s; on the contrary, the latter were sampled independently from the correct distributions but they do not provide information on the rank correlation among the variables.

To obtain a dataset with both the mentioned properties, the values of d*, v* and s* were then ranked and replaced with the corresponding percentiles derived from the ordered versions of d, v and s."

**R2.C10**. Table 2: Define "ECDF" in the table or provide the abbreviation at the end of the table for better understanding.
**Reply**: We will include the definition of "ECDF" (empirical cumulative distribution function) at the end of the table to enhance clarity.

**R2.C11**. Line 174 and 177: Clarify the difference between hypothetically flooded buildings and the generated synthetic dataset. Consider listing the involved datasets in a table or figure and explaining their significance.
**Reply**: As in the general comment raised by the Reviewer, we recognize the need for clarity regarding the various datasets used in this study, which will be handled in the revised manuscript, by explicitly distinguishing and explaining the significance of each dataset involved in the study. Specifically, concerning lines 174 and 177, we underline that the 5000 hypothetically flooded buildings are directly sampled from the synthetic dataset, the characteristics of which are detailed in Tables 1 and 2.

**R2.C12**. Figure 3: Provide further explanation for the labels in the figure.
**Reply**: The figure labels represent various building parameters such as footprint area (FA), basement area (BA), external perimeter (EP), internal perimeter (IP), and basement perimeter (BP). The explanations for each label are already shown in the original figure caption.

**R2.C13**. Line 235: Explain the diagonal axis of Figure 2, including the value of the y-axis.
**Reply**: The diagonal of Figures 2 and 3 shows the density plots for the different variables; in the revised version of the manuscript we will amend the figures by fixing the label of the y-axis for the plots on the diagonal by reporting density values.

**R2.C14**. Line 255: In Figure 4, clarify the meaning of the bar colors and the red dots. Additionally, explain the color progression from dark blue to light blue and indicate if it represents a rank sequence. Complete all figure labels or provide the necessary information in the figure caption.
**Reply**: In Figure 4, the bar colors and color progression represent a rank sequence based on the median value of the damage difference, with darker blues indicating higher variable importance. This ranking information was already outlined in the original figure caption. The red dots within the figure represent outlier values, a detail that will be explicitly included in the revised figure caption for a better clarity.

**R2.C15**. Figure 7: Increase the resolution of the figure.
**Reply**: The figure is originally generated in high resolution (600 DPI). The observed issue with resolution is likely related to the rendering during the generation of the PDF file.

---

## Author Response (AR1)

We would like to thank the Reviewer for his interest in our work and for carefully reading our manuscript; we greatly appreciate the insightful comments as they contribute to increase the manuscript robustness and, in general, to improve its quality. We provide below a point-by-point reply to the general and specific comments raised.

You can see all the changes made on the manuscript in the file "Di_Bacco_et_al_rev_track_changes". The "clear" version of the revised manuscript is in the file "Di_Bacco_et_al_rev".
* * *
**REVIEWER 1**

**Summary**

The manuscript "The value of ultra-detailed survey data for an improved flood damage modelling with explicit input data uncertainty treatment: INSYDE 2.0" proposes a tailored flood impact modeling framework INSYDE to account for the lack of information/uncertainty with regards to the required micro-scale vulnerability and exposure characterstics of buildings. In this study, the framework and process of data-preparation are discussed alongside three test cases to show the benefits.

The authors have done a good job. The authors are addressing a very relevant need for better fit tools to support decision-making acknowledging uncertainty. The authors offer a comprehensive idea how to deal with limited knowledge and give insights into the sensitivities. The manuscript is generally well written and makes particular good use of Tables. The overall structure of the manuscript could be improved alongside revising some of the figures and reflecting on their use. Strengthening the discussion of the benefits and learnings for a decision-maker from using INSYIDE 2.0 compared to other models, could make this manuscript a very strong and relevant addition to the scientific community.

**General comments**

**R1.C1.** The authors do a great job in justifying the two research questions to explore in this study. While this reviewer can clearly identify the evidence presented to answer the second research question. However, the first question on arguing the added value in terms of output quality and usefulness seems to be addressed marginally and should receive more attention in the results and discussion section. What can decision-makers specifically learn from INSYDE 2.0? In that context, this reviewer observed that the Authors seem not to discuss any limitations of this approach.

Reply: The primary lesson learned from INSYDE 2.0 lies in transcending the confines of deterministic damage models. By challenging the conventional notion of certainty in damage estimations, our approach aims at highlighting (i) the importance of recognizing uncertainty and (ii) the idea, from a decision-maker perspective, that the band of uncertainty can often be more informative than a singular point estimate (which should not be regarded as a definitive "truth", but subject to uncertainty). In our paper we also challenge the conventional use of the term "validation" in the context of flood damage modelling. Acknowledging the complex interplay of assumptions in both model input and output, we refrain from using the term "validation", as it may imply a level of certainty that is inherently elusive. Our approach, considering the uncertainties in input data and recognizing possible biases in observed damage, shifts from just seeking convergence between estimations and observations to embracing a comprehensive understanding of the uncertainties that characterize flood damage estimations. This points have been introduced in the revised version of the conclusions at P14.L392-401.

Inherent to the multivariable nature of the model, INSYDE 2.0 demands a wealth of detailed, locally dependent input data. This characteristic, while enhancing the model's granularity, simultaneously presents a challenge in data acquisition, as already highlighted in multiple parts of the original manuscript ("While data retrieval on large-scale for some of the vulnerability variables can be costly" / "even though such tasks can be time-consuming, especially in consideration of the possible significant regional spatial variability of the building stock"). In addition, in the updated version of the paper (P15.L402-403), we have also mentioned another possible limitation of the presented study, concerning the necessity, for The Po river application, of having introduced certain assumptions on some variables, due to the lack of specific information on them to derive representative distributions.

**R1.C2.** Line 95: This reviewer thinks that the description of the methodological approach could benefit from improved visualization and explanation as well as restructuring the sections. Figure 1 seems complete but

complex. The reviewer cannot recognize the elements mentioned in the figure in the accompanying text (or the subsequent subsection titles). This makes it very difficult to follow. Using more descriptive subsection titles aligned with the steps in an (updated?) Figure 1 could avoid this challenge. If others were to use INSYDE 2.0, would they use this methodological approach as well? If so, making a clear distinction between describing INSYDE 2.0, Preparing data for INSYDE 2.0 and Applying INSIYDE 2.0 could be beneficial in the method section.

**Reply**: To enhance the understanding of our approach, we have revised Figure 1 to better align with the accompanying text and make it more comprehensible. In particular, we have better highlighted the different methodological steps of our study, with the identification of the three main blocks of the analysis, which now also match with the titles of the different subsections of the manuscript.

**R1.C3.** This reviewer is wondering whether the title (particularly: 'The value of ultra-detailed survey data') of this manuscript captures the main purpose of this study. Firstly, because the data-sources discussed in this paper are not limited to survey data. Secondly, the paper mostly focuses on exploring the effects of uncertainty.

**Reply**: We agree with the Reviewer that our manuscript includes a broader range of data sources beyond survey data. To accurately reflect the diversity of data employed in our study, we have then modified the first part of the title to "The value of multi-source data", while we believe the second part of the original title already effectively communicated the focus around exploring the impacts of uncertainty in flood damage modelling. Then, the new proposed version of the title is the following: "The value of multi-source data for an improved flood damage modelling with explicit uncertainty treatment: INSYDE 2.0"

**Specific comments**

**R1.C4.** Line 23: these references seem outdated to confirm the Author's claim regarding development and remaining limitations of flood risk modelling over the past decade.

**Reply**: In the revised version of the manuscript, we have incorporated more recent references (including Wagenaar et al. (2016) (already present in the reference list of the original manuscript), Winter et al. (2018) and Marvi (2020)) to provide a current and comprehensive overview on current limitations of flood risk modelling.

*Winter, B., Schneeberger, K., Huttenlau, M. and Stötter, J.: Sources of uncertainty in a probabilistic flood risk model. Nat Hazards **91**, 431-446 (2018). doi: 10.1007/s11069-017-3135-5.*

*Marvi, M.T. A review of flood damage analysis for a building structure and contents. Nat Hazards, **102(3)**, 967-995. (2020). doi: 10.1007/s11069-020-03941-w.*

**R1.C5.** Line 103 – Line 106: What benefits did the Authors see in using data from a specific region to explore the sensitivities of the impacts/feature importance? Would the findings from a sensitivity analysis not offer similar insights, perhaps even more generalizable?

**Reply**: The rationale for our approach is rooted in the need for context-specific insights into flood damage assessment. By analyzing a specific region, it is possible to identify the input variables that have the most significant impact on damage assessment in that particular context. This region-specific understanding is crucial for guiding efficient data retrieval efforts, as it allows to prioritize the collection of information on variables that really play a key role in the given region (i.e., issue of transferability of damage models, which should be considered as "local" models).

In our analysis, we therefore performed a sensitivity analysis for two different scenarios to provide a comprehensive perspective. The first involved an analysis tailored to the area under investigation, with typical ranges of values calibrated based on observed data for the Po River region. This allowed us to discern the most important variables for damage estimation in the specific context (top panel of the original Figure 4). The second scenario considered a non-region-specific case, encompassing a broader range of values for the input variables, to examine general sensitivities of damage (bottom panel of the original Figure 4).

We believe that this dual approach provides both essential insights for effective and efficient damage modelling at the regional level and broader, more generalizable findings on damage mechanisms. This aspect has been clarified in the revised Section 2 of the manuscript and, more specifically, at P5.L172-176.

**R1.C6.** Line 108 – Line 119: Did the authors generate the EDF's? How many data points were available for fitting the distributions mentioned in Tab.1 and Tab.2 ? Since this study is addressing effects of uncertainty, it would be interesting to know on what basis these distributions are developed . This reviewer thinks it could be a good idea to add these EDF's + fits in the Supplemental Material.

**Reply**: As mentioned in the original manuscript, details on data statistics at the base of derived distributions for the Po River are available in the work by Huayra Mena (2022), which reports detailed information on building inventory, virtual surveys and flood-related data.

Due to the public nature of the mentioned document, we cannot reproduce the same information here (i.e., possible plagiarism issue), but we have enhanced the description of the data sources in the revised version of the manuscript to provide more details on the data sources from which we derived the empirical distributions (as in the revised version of Section 2.2, P4.L120-144).

**R1.C7.** Line 120: How were different return periods included in the EDF's?

**Reply**: The available hazard maps for the Po River included information on three different return periods. However, for the derivation of the EDFs, our focus was specifically on the most representative medium-frequency scenario (i.e., typical design scenario for the implementation of mitigation measures in the Po District). We have clarified this aspect in the revised version of the manuscript (P4.L134-146).

**R1.C8.** Meanwhile the current text in section 2.3 is interesting and probably relevant for the functioning of the model, this reviewer does not see the direct link between the purpose of this study (include uncertainty into INSYDE) with this fix addressing the scalability problem. Since this bias seems to be mentioned in 2.1 (line 83), this reviewer would suggest to mention the change of the model in 2.1 and/or put the detailed elaboration in the Supplementary Material. Given that the change already had been applied when using INSYDE in Belgium, this seems not innovative or relevant to report here. Instead, this reviewer was expecting an elaboration of the statement in line 100 elaborating on the process of translating mixed source data into distributions or required adjustments to the (existing) probabilistic framework.

**Reply**: Following Reviewer's suggestion, we have merged the original section 2.3 into section 2.1. Furthermore, as also pointed out in our reply to R1.C6, in the revised version of the manuscript we have now included more details on the data sources used for the derivation of the empirical distributions for the Po River District.

**R1.C9.** Line 148 – line 150: Why did the authors choose to build one dataset containing two different flood types (riverine and flash floods)? Would it not be more accurate to have two separate ones, one per flood type?

**Reply**: The rationale behind constructing a dataset encompassing both riverine and flash floods was to establish a single model for the entire Po district, aligning with exposure and vulnerability characteristics applicable to the whole region (i.e., although different areas within the district may encounter distinct flood types, the overall exposure and vulnerability features are the same for the entire region). This has been clarified at P4.L136-138 of the revised version of the manuscript.

Furthermore, it is important to note that the fundamental feature of INSYDE, as a local model, is flexibility. Users can indeed modify and fine-tune information related to flood features and building characteristics as per their specific needs and preferences. This adaptability ensures the applicability of the model to diverse scenarios, as emphasized within the manuscript.

**R1.C10.** Line 152: These distributions are generated based on the Po river modelling exercise only? Or also for the two historic case studies separately?

**Reply**: The generated distributions are applicable to the two historic cases, both situated within the Po plain. However, for these specific flood events, detailed information on water depth and flow velocity at the building location and a rough estimate of flood duration were available (see Table 3 in the original manuscript) and it was not required a sampling from the elaborated distribution (except for the sediment variable).

**R1.C11.** Line 155 – Line 163: The study would benefit from additional elaboration how these correlations are built. This reviewer can understand why authors built a synthetic correlation between inundation depth and duration (does it work differently for riverine floods and flash floods?), but much less with regards to the flow speed and flood duration. Can the Authors elaborate on these decisions? How accurate was the fitting of the correlation function based on the sample values? In case of the Po river modelling, flow depth and flow velocity were modelled and thus directly correlated already? What does d / max(d) stand for? Why did the Authors not use Copulas to account for joint probabilities?

**Reply**: For the analyzed area, while we had information for deriving distributions for water depth and flow velocity, the same cannot be said for sediment and duration. Therefore, as already stated in the original manuscript, synthetic correlations were established among the hazard variables, adopting an expert-based, physically inspired approach to represent different flow types. For instance, high flow velocity is considered to be related with increased sediment-carrying capability, as opposed to long-lasting riverine floods that

typically carry fine-graded sediments. These patterns are translated into the functional forms presented in L156-158 of the original paper and graphically shown in Figure 2. The expression d / max(d) is employed to prevent the occurrence of negative velocity values. As for copulas, while they may be useful for modelling joint probabilities, the simplicity of our synthetic correlations, guided by physical principles, was deemed sufficient for our purposes.

**R1.C12.** Line 163 – Line 165: This is not clear. It seems that he and v are the leading parameters and d, s, q are depending on these parameters. Why do we lose information? And what can this reviewer picture under "[…] the values of d\*, v\* and s\* were then replaced with the correspondent percentiles from the datasets of d, v and s"? What is the effect of this?

**Reply**: In the revised version of the manuscript, we have clarified the procedure by amending the related text as follows:

"More in detail, probability distributions were first retrieved independently for the hazard variables based on detailed data when available (he, v) or upon expert-based assumptions derived from aggregated or approximated data (d, s, q), and used to sample sets of 250.000 elements; furthermore, the following functional dependencies were assumed to describe the correlation among the features, based on the values sampled for he, d and v:

$$d^* = c_1 + c_2 \cdot \sqrt{he} \cdot N(\mu = 1, \ \sigma = 0.2)$$

$$v^* = c_3 - d/\max(d) \cdot N(\mu = 1, \ \sigma = c_3 - d/\max(d))$$

$$s^* = c_4 + c_5 \cdot \sqrt{v} \cdot N(\mu = 1, \quad \sigma = 0.2)$$

with N being a random number from a normal distribution with mean $\mu$ and standard deviation $\sigma$, while the coefficients $c_i$ are constant values introduced in the expert-based approach to obtain the desired functional relationships among the variables. q was instead assumed independent from the other hazard features.

Although the resulting d\*, v\* and s\* account for the correlation among the hazard variables, they do not follow the probability distributions retrieved independently for the variables d, v and s; on the contrary, the latter were sampled independently from the correct distributions but they do not provide information on the rank correlation among the variables.

To obtain a dataset with both the mentioned properties, the values of d\*, v\* and s\* were then ranked and replaced with the corresponding percentiles derived from the ordered versions of d, v and s."

**R1.C13.** Line 174: How did the authors end up with the number of 5000 hypothetical buildings? Did the authors explore the convergence behavior? When looking into uncertainty and Monte Carlo sampling justification of such choices should be provided.

**Reply**: Upon revision of the manuscript, to be more comprehensive, we decided to run the feature importance analysis for the whole synthetic dataset consisting of 250,000 elements, obtaining consistent results with the previous run on the smaller sample (noticing only very minor differences). Related text in the revised manuscript has been updated accordingly.

Also based on this result, for the second test, involving the absence of multiple inputs, we opted for maintaining the smaller sample of 5,000 elements, which ensured a trade-off between the robustness of the results and computational efficiency (since for each building we run 1,000 replicates, resulting in a total of $5 \cdot 10^6$ runs for damage estimations).

**R1.C14.** Line 179: In the results, readers are presented with the damage difference as the metric to explore the feature importance. Information on how this metric is calculated (e.g. aggregated vs averaged over the 5000 buildings) would clarify how to read the results.

**Reply**: As already described in the original manuscript, the metric used for the feature importance is obtained through a probabilistic test on the synthetic dataset (considering 250,000 elements - in the revised version of the manuscript, as described in our reply to R1.C13). Initially, we use INSYDE 2.0 to estimate damage on the complete dataset, assuming all input values are available, establishing a reference point. Subsequently, for each of the 250,000 buildings, we systematically remove one input variable at a time, sampling the corresponding missing values from the synthetic dataset. This process is repeated for each variable, and each time, damage is recalculated. The absolute difference in damage compared to the reference value is then recorded, allowing us to determine the variance induced by each feature on model outcome. Hence, the boxplots in Figure 4 depict the range of damage differences observed across the 250,000 tested buildings for

each examined missing variable. The representation of data variability by means of boxplot is a standard representation and, in our opinion, does not deserve further explanations in addition to the procedural description already provided in the original manuscript.

**R1.C15.** Line 195: per house 1000 replicates were generated to account for uncertain combination of ~ 20 parameters. Did the authors explore the convergence behavior of the results to confirm that this choice is reliable (see comment regarding line 174)?
**Reply**: See response to comment R1.C13.

**R1.C16.** Line 215 – Line 220: Here the Authors mention INSYDE and its benefits. Showcasing the utility of INSYDE 2.0 would not only be towards a decision-maker but also towards the previous version INSYDE. It would be interesting to see the results of the old INSYDE alongside INSYDE 2.0.
**Reply**: The evolution of INSYDE from its original version has been marked by ongoing enhancements, refinements and applications. Results for the two case studies with the older INSYDE versions, which utilized fixed default values in the case of missing inputs, have been extensively presented in prior works, in which modelling outcomes have been compared against observed damage. In the revised version of the manuscript, we have provided more detailed reference and discussion to these results at P14.L366-368. However, it is worth noting that the central point of the present study lies not only in showcasing the incremental improvements from one INSYDE version to another but, more importantly, in underlining the critical awareness of the influence of uncertainties stemming from unknown input data in model applications. While the different versions of INSYDE could certainly be informative, our primary emphasis here is on the broader message, i.e. on the significance of acknowledging uncertainties in the modelling process. This awareness stands as a substantial added value, crucial for both the modeler and the decision-maker, surpassing any potentially misleading sense of certainty derived from a deterministic application of (any) model. This point has been better highlighted in the revised version of the manuscript at P14.L368-373 and P14-15.L393-401.

**R1.C17.** Line 220: Table 3 is very helpful. This reviewer would suggest to add the details regarding the synthetic case study in that table as well for overview purposes.
**Reply**: In the synthetic case study with the reduction in the dataset's level of completeness, with the exception of FA and he, all other input variables were assumed to be possibly unknown and, therefore, we think that providing additional details in Table 3 might not be meaningful. Comprehensive information regarding this aspect is instead already available in Section 2.5.2.1 of the original manuscript.

**R1.C18.** Line 224: The benefit of section 3.1 is not clear to this reviewer. It seems to focus on the pairwise occurrence of parameters. It is unclear how it offers evidence to answer the initially proposed research questions. This reviewer thus suggests to either incorporate it in the methodology section or place it in the Supplemental Material. For this reviewer, pairwise occurrence is just one of the different elements in the data sampling process. For example, the pairwise occurrence in the collected information used for the sampling might be of additional interest (to gain insight into uncertainty progression). At the same time, Figures 2 and 3 seem to have an incorrect design: the subplots on the diagonal seem to be histograms, but the y-axis labels are not correct.
**Reply**: Section 3.1 visually represents the identified distributions and the pairwise dependencies among the variables, as outlined in Tables 1 and 2 (please refer to the original Section 2.4, L147-164). While we may understand the suggestion to relocate it, these graphical representations serve as informative and explicative visualizations of the methodological approach for the generation of the synthetic dataset, which is used in the downstream applications presented in the subsequent sections of the paper.
We have instead amended Figures 2 and 3 by correcting the y-axis for the plots on the diagonal (density plots) and simplifying the figures by displaying only the lower diagonal part of the pair plot (as it is a symmetrical matrix).

**R1.C19.** Line 253: Here, an extended dataset is mentioned the first time (Elaboration in the Method section needed!). So authors are using 4 case studies? In general, what is the justification for the three different cases? What added benefit do the authors see by adding a second stylized case here?
**Reply**: We introduced a case based on an extended synthetic dataset with a broader range of variability for the input variables in order to offer insights into general damage sensitivities to input variables beyond the specific case of northern Italy. While the regional case highlights the crucial variables for efficient damage modelling in the area under investigation, the more general case provides broader information on the influence of different variables on estimated damage. This clarification, along with the related adjustments in the methodological section, has been incorporated into the revised manuscript (see also reply to comment R1.C5).

**R1.C20.** Figure 4: How is the damage difference calculated? The medians are barely visible. In general this figure is very colorful, while some other elements are not visible (whiskers, box whiskers for LM to YY in upper plot). The bars seem to go beyond the chosen y-axis limits on some occasions , e.g for BE (upper plot).
**Reply**: The method for calculating the (absolute - this has been clarified in the updated version of the manuscript) damage difference, serving as the metric for the feature importance, is described in detail in Section 2.5.1 of the original manuscript (see also our response to Comment R1.C14). In the updated paper, we have improved Figure 4, by better visualizing its crucial elements. However, we highlight that the y-axis origin is set at 10€ to neglect not significant damage differences, while maintaining figure readability.

**R1.C21.** Line 260: how do the correlations between he, d, v, and s play into these results?
**Reply**: The correlations established between the hazard variables are crucial for generating representative and meaningful data. Without these correlations, there would be a risk of generating "non-physical" data, resulting in unrealistic scenarios (e.g., associating a 0.01 cm inundation depth with a 48-hour duration), which would significantly impact the accuracy and representativeness of damage estimation. As a consequence, the variability in the damage difference is reduced, since we are avoiding the effect of such "non-physical" input combinations.

**R1.C22.** Line 261: 10,000 EUR per house or averaged across the entire set of buildings? Are these higher damages or lower damages? A permutation of only 10,000 EUR for each house would lead to a difference of up to 5 million EUR, which is significant again? What are the uncertainty bounds for this difference?
**Reply**: The value of 10,000 € represents the median of the absolute damage difference calculated across the synthetic building portfolio and the boxplot in Figure 4 directly shows the uncertainty bounds associated with the damage differences (see also reply to comments R1.C14 and C20.). In the revised version of the manuscript, we have modified the y-axis label by reporting "Absolute damage difference [€]" to enhance the clarity of the figure.
While the amount of 10,000 € may seem high, it is not surprising, since it is associated with (possible lack of) information on inundation depth. If, for a particular building, there is no available information on inundation depth, the model must sample a value, which can be very different from the one for which the reference damage has been calculated. This effect is more pronounced in the extended dataset scenario, where a broader range of values is possible for water depth, resulting in larger damage differences.
However, it should be recognized that in practical applications of damage assessment, inundation depth is a mandatory and usually known variable (or at least known within certain limited uncertainty). This is because an inundation scenario must be defined. As a result, the uncertainty associated with this variable is more effectively constrained within narrower bounds compared to scenarios where inundation depth is randomly sampled (which would mean performing a damage assessment for an unknown inundation scenario).

**R1.C23.** Figure 5: The left plot is very useful and clear to see. Why did the Authors choose to combine the scatter plots for the other BT's?
**Reply**: The decision to combine the scatter plots for the other BTs in Figure 5 was guided by the similarity in the patterns among BTs, allowing for an efficient visualization; the use of distinct colors for each BT facilitates the clear identification of individual patterns, while the subdivision into a separate plot for the apartment type was necessitated by the different scale of the y-axis for this BT.

**R1.C24.** Figure 7: While Table 4 is very helpful and supports the reasoning in the text, this reviewer has doubts regarding Figure 7. First of all, the use of log-log scales makes it very difficult to interpret the results since distance is not constant at different positions in the figure. As such, diverging from the diagonal has much more severe implications towards increasing Observed Damages. Secondly, this reviewer is wondering whether making use of the damage difference as in previous figures might be more informative and supportive regarding the research questions. What is a decision-maker learning from this visualization about uncertainty in modelling?
**Reply**: The use of log-log scales was intentional to allow a better visualization of the full range of data; otherwise, the lower values would have appeared as collapsed to a single region of the plot. In addition, we preferred to maintain the representation of results in terms of observed vs calculated damages (and not in terms of damage difference), as conventional in validation exercises.
According to us, the visualization of error bars is crucial for decision-makers, providing insights into the associated uncertainties in a damage model. This aligns with our broader goal of emphasizing the importance of acknowledging and addressing modelling uncertainties, avoiding the potential pitfalls of deterministic estimations that might give a misleading sense of certainty. Furthermore, the visualization of uncertainty,

seldom provided in usual "validation" exercises of damage models in the literature, adds value to our proposed approach.

**R1.C25.** Table 4: What is the computed expected damage? Other point: As a decision-maker, a conclusion I would draw form this Table is that Lodi and Caldogno are both cities that are on average much more vulnerable (because of building properties) than other cities in Northern Italy. Or that INSYDE underestimates damages. Can a decision-maker get any insights into how much of the model gap with reality can be attributed to the uncertain input data vs. other sources of uncertainty/error (e.g. the hazard-damage relations of the model). What lessons can a decision-maker learn from this? Elaborating a bit more on this could benefit the significance of this work.

**Reply**: Theoretically, the computed expected damage for each building would be the average of the damage outputs for an infinite number of runs. In our opinion, from a decision-makers' perspective, the entire range of damage variability (Table 4) is more informative than a single value.

It appears that the comment from the Reviewer originates from the assumption that "observed damage" represent an absolute truth. However, as highlighted in Molinari et al. (2020), observed damage does not always fully capture the reality, being subject to various biases. This is also the reason why we prefer to avoid the use of the word "validation". Consequently, drawing concrete conclusions about model's underestimation or overestimation becomes challenging. Instead, from a decision-maker's perspective, the key message would be about the transparency on the uncertainty in the damage estimation. Rather than focusing on a single value like the expected damage, decision-makers should be aware of the broad variability of model outcomes stemming from limited knowledge on certain inputs. To make well-informed decisions, a comprehensive understanding of modelling limitations and assumption is essential. This consideration was already present in the original manuscript, but we have further emphasized it in the revised version of the conclusion section (see also reply to comment R1.C1).

**R1.C26.** Line 350-352: This reviewer has not seen any evidence that confirms this claim. The advantage can only be compared to the original INSYDE set up, or the learnings of analysing the uncertainty bands.

**Reply**: The evidence supporting the claim is present in Table 4, where observed damage consistently falls within the 50[th] and 75[th] percentiles of calculated damage for both case studies. Additionally, to provide a more comprehensive view, we have included reference and discussion from previous studies that report "validation results" for the two case studies using earlier versions of INSYDE. See also reply to comments R1.C16 and C25.

We would like to thank the Reviewer for his/her interest in our work and for carefully reading our manuscript; we greatly appreciate the insightful comments as they contribute to increase the manuscript robustness and, in general, to improve its quality. In the following, we provide a point-by-point reply to the general and specific comments raised.
* * *
**REVIEWER 2**

The paper introduces INSYDE 2.0, a flood damage modeling tool that integrates ultra-detailed survey and desk-based data to enhance the reliability and informativeness of flood damage predictions. By incorporating a probabilistic module, it addresses missing input data and provides transparent information about uncertainties arising from limited knowledge of damage explicative variables. This integration significantly improves the reliability and robustness of flood damage assessments. While the extensive use of ultra-detailed data contributes to the model's reliability and the importance of accurate damage estimation, the logical coherence and readability of the manuscript still require significant improvement. Therefore, a major revision is necessary to strengthen the manuscript for potential publication.

**Major comments**:

**R2.C1.** One of my main comments is regarding the dataset used in this study. The manuscript introduces various datasets, such as synthetic datasets, observed datasets, survey datasets, auxiliary datasets, specific datasets, and empirical datasets. However, it is unclear what each dataset represents and how they are related. Are these datasets intended to establish the functional relationships between building parameters, serve as reference data, or facilitate model validation? The lack of clarity regarding the nature and relationships of these datasets, as well as their correspondence with the terms used in Figure 1, significantly hampers readability. I think it's good to provide a bit more information on the construction of all these datasets. Particularly since this manuscript focuses on uncertainty analysis.
**Reply**: We agree with the Reviewer that the original description of the various datasets involved in our study could have lead to confusion. The revised manuscript, by maintaining a consistent terminology throughout the text and with the support of the updated version of Figure 1, now better describes the purpose of each dataset and how they are interrelated.

**R2.C2.** The description of building damage in the paper is not very intuitive, and the location distribution information of buildings is hardly mentioned, so it is difficult for readers to apply INSYDE 2.0 to their examples. In my opinion this question - how do we sample complex and spatially distributed variables in a meaningful way? - is one of the key research questions that the uncertainty and sensitive analysis community will need to work on if we want to move on to the next stage of applying this type of technique to complex models. I am not suggesting the authors should solve this issue, but I think they should point out this is a very important and critical step and a big area for future research and development.
**Reply**: In the revised manuscript, we have included some additional information on the study area, including the incorporation of a map to provide a clearer understanding of the geographical context (Figure S1 in the Supplementary material and text at P4.131-134). We have not included the map in the main text (but only in the Supplement) and we have not provided an ultra-detailed description of the study area, since the Po River District has been used only as an exemplificatory case of the application of the methodological framework, that can be theoretically replicated also elsewhere, upon suitable adaptation. We acknowledge that addressing the question of how to effectively sample input variables for damage models is a key challenge and an area for future research and development within the community. In our paper we have exactly proposed an attempt for solving this issue (with a framework that can be replicated in different areas), which consists in deriving representative distributions from multi-source data, which allow to sample input features at the local scale in case of unknown variables for model application.

**R2.C3**. In Section 2.2, the meaning of "specific catchments of the district representing the distinctive inundation types occurring in rural or urban areas as well as in flat or steeper regions within the Po River district" is unclear. It would be beneficial to include a spatial distribution map or a dem illustrating the Po River district to provide a better understanding of the study area.

**Reply**: A map of the Po River District has been incorporated in the revised version of the Supplementary Material to provide a visual representation of the study area (see also reply to R2.C2).

**R2.C4**. While it is described that nearly 5000 hypothetically flooded buildings are analyzed, the manuscript does little to explain or describe the building characteristics involved. Numerous contributing factors affect economic loss. Given the primary objective of this manuscript is to quantify loss, it would be necessary for the authors to describe the 5000 hypothetically flooded buildings in more detail.
**Reply**: In the revised manuscript, we have made clear that the (not more 5,000 – see reply to R1.C13) 250,000 hypothetically flooded buildings are sampled from the distributions generated for the Po River District, whose characteristics are already comprehensively described in the original Table 2.

**R2.C5**. In lines 298-301, I guess the authors mean that high variability in certain parameters leads to significant deviations in loss estimation. If so, this issue represents a substantial portion of the cases and should be addressed. The authors may consider discussing potential measures to reduce such errors.
**Reply**: In the specified lines, the text indeed addressed the fact that high variability in certain variables can lead to significant deviations in loss estimation. This issue was thoroughly explored in the section discussing the feature importance, where it was acknowledged that certain variables carry more importance than others due to their role played on damage mechanisms. Additionally, another aspect is also discussed: it points out that variations in input parameters may only result in damage variations if certain conditions on other parameters are met.
Regarding the suggestion to identify measures to reduce "errors," it is important to note that these are not errors, but uncertainties. Instead of just aiming to reduce uncertainty, the focus of our study is on providing explicit information about it and highlighting the importance of the features that contribute to this uncertainty. This perspective has been maintained in the revised version of the manuscript.

**Minor comments**:

**R2.C6**. Language and Sentence Structure: Please carefully review the manuscript for sentence clarity and readability. Some long sentences, such as lines 301-303, are difficult to follow. Consider breaking them into two sentences or rewriting them for better readability.
**Reply**: In revising the manuscript, we have enhanced the overall readability of the text, by paying particular attention to longer sentences to make them clearer.

**R2.C7**. Line 154: Typo - "250.000" should be corrected.
**Reply**: Fixed (250,000).

**R2.C8**. Line 156-158: Please describe the meaning of values "$c_1$-$c_5$."
**Reply**: They are constant values introduced in the expert-based approach to obtain the desired functional relationships among the variables. We acknowledge that this may not have been explicitly mentioned in the original manuscript and we have therefore clarified it in the revised version of the paper (see reply to the following comment).

**R2.C9**. Line 163-164: Clarify the meaning of "to obtain the correct distributions without losing information on the interdependence among the variables, the values of d*, v*, and s* were then replaced with the correspondent percentiles from the datasets of d, v, and s."
**Reply**: This sentence has been improved for clarity in the revised version of the manuscript as follows:
"[…] More in detail, probability distributions were first retrieved independently for the hazard variables based on detailed data when available (he, v) or upon expert-based assumptions derived from aggregated or approximated data (d, s, q), and used to sample sets of 250,000 elements; furthermore, the following functional dependencies were assumed to describe the correlation among the features, based on the values sampled for he, d and v:

$$d^* = c_1 + c_2 \cdot \sqrt{he} \cdot N(\mu = 1, \ \sigma = 0.2)$$

$$v^* = c_3 - d/\max{(d)} \cdot N(\mu = 1, \ \sigma = c_3 - d/\max{(d)})$$

$$s^* = c_4 + c_5 \cdot \sqrt{v} \cdot N(\mu = 1, \qquad \sigma = 0.2)$$

with N being a random number from a normal distribution with mean μ and standard deviation σ, while the coefficients $c_i$ are constant values introduced in the expert-based approach to obtain the desired functional relationships among the variables. q was instead assumed independent from the other hazard features.

Although the resulting d*, v* and s* account for the correlation among the hazard variables, they do not follow the probability distributions retrieved independently for the variables d, v and s; on the contrary, the latter were sampled independently from the correct distributions but they do not provide information on the rank correlation among the variables.

To obtain a dataset with both the mentioned properties, the values of d*, v* and s* were then ranked and replaced with the corresponding percentiles derived from the ordered versions of d, v and s."

**R2.C10**. Table 2: Define "ECDF" in the table or provide the abbreviation at the end of the table for better understanding.

**Reply**: For more clarity, in the revised version of Table 2 we have used a more explicit text for referring to the different empirical distributions (see revised version of Table 2).

**R2.C11**. Line 174 and 177: Clarify the difference between hypothetically flooded buildings and the generated synthetic dataset. Consider listing the involved datasets in a table or figure and explaining their significance.

**Reply**: As in the general comment raised by the Reviewer, we recognized the need for clarity regarding the various datasets used in this study, which has been addressed in the revised version of the manuscript. Specifically, concerning lines 174 and 177, we underline that the (not more 5,000 – see reply to R1.C13 and R2.C4) 250,000 hypothetically flooded buildings are directly sampled from the synthetic datasets (for the Po District and for the extended case – see also reply to comment R1.C5), the characteristics of which are detailed in Tables 1-2, as well as in Figures 2-3 and S2-S3.

**R2.C12**. Figure 3: Provide further explanation for the labels in the figure.

**Reply**: The figure labels represent various building variables, such as footprint area (FA), basement area (BA), external perimeter (EP), internal perimeter (IP), and basement perimeter (BP). The explanations for each label are already reported in the original figure caption.

**R2.C13**. Line 235: Explain the diagonal axis of Figure 2, including the value of the y-axis.

**Reply**: The diagonal of Figures 2 and 3 shows the density plots for the different variables; in the revised version of the manuscript we have updated the figures by amending the label of the y-axis for the plots on the diagonal by reporting density values.

**R2.C14**. Line 255: In Figure 4, clarify the meaning of the bar colors and the red dots. Additionally, explain the color progression from dark blue to light blue and indicate if it represents a rank sequence. Complete all figure labels or provide the necessary information in the figure caption.

**Reply**: In Figure 4, the bar colors and color progression represent a rank sequence based on the median value of the damage difference, with darker blues indicating higher variable importance. This ranking information was already outlined in the original figure caption. The red dots within the figure represent outlier values, a detail that has been explicitly included in the revised figure caption for a better clarity.

**R2.C15**. Figure 7: Increase the resolution of the figure.

**Reply**: The figure has been originally generated in high resolution (600 DPI). The observed issue with resolution is likely related to the rendering during the generation of the PDF file.